# Use of a Unique Mobile Medical Asset in COVID Monoclonal Antibody Treatment

**DOI:** 10.3390/healthcare9080990

**Published:** 2021-08-04

**Authors:** Herman Morchel, David Clark, Leighanne Buenvenida, Chinwe Ogedegbe

**Affiliations:** 1Hackensack University Medical Center, Hackensack Meridian School of Medicine—Hackensack Meridian Health, Hackensack, NJ 07601, USA; chinwe.ogedegbe@hmhn.org; 2Jersey Shore University Medical Center—Hackensack Meridian Health, Neptune, NJ 07753, USA; davidc.clark@hmhn.org (D.C.); leighanne.buenvenida@hmhn.org (L.B.)

**Keywords:** COVID-19, Mobile Satellite Emergency Department, COVID monoclonal antibody treatment, Bamlanivimab, COVID-19 patient isolation, COVID-19 patient treatment area

## Abstract

The COVID-19 pandemic and the subsequent surge of patients presented to emergency departments has forever changed the paradigm of delivering emergency care. The highly infectious nature of the 2019 Novel Coronavirus, or COVID-19, mandated strict environmental changes, novel patient care, and flexible strategies to continue to deliver efficient emergency care while maintaining appropriate physical distancing between suspect and non-suspect COVID-19 patients. The engagement of a unique rapidly deployable Mobile Satellite Emergency Department (MSED) with scalable capability from prompt care to resuscitation level allowed the emergency care team to optimize patient care and throughput. The MSED was strategically located adjacent to the ambulance entrance. While initially deployed to increase Emergency Department surge capacity, the MSED was repurposed to cohort and treat COVID patients with the monoclonal antibody, Bamlanivimab, who were expected to be discharged after treatment. This allowed for more efficient use of Emergency Department resources, including physical space and staffing.

## 1. Introduction

This communication describes the use of a unique mobile medical asset to improve patient care and optimize the use of Emergency Department (ED) resources during the COVID-19 pandemic.

The location of use is the Hackensack Meridian Health—Jersey Shore University Medical Center in Neptune, New Jersey. The Jersey Shore University Medical Center is a 630-bed academic medical center in the Eastern Central part of the state. The Emergency Department volume is approximately 88,300 patients per year, with 20,000 being pediatric (2019 pre-COVID).

The specific innovation addressed in this communication is the treatment of patients diagnosed with COVID-19 with the monoclonal antibody Bamlanivimab in a unique alternate care area. These are generally patients who are scheduled to present for treatment and whose discharge is anticipated after treatment. The process can take several hours, since it involves evaluation, intravenous infusion, and re-evaluation post-treatment. Coupled with the required isolation, a significant amount of Emergency Department resources can be consumed.

The availability of a unique trailer based mobile medical asset, coupled with patients being scheduled well in advance, has allowed the implementation of a specialized patient flow pathway to optimize care and throughput.

Patients receiving the antibody were treated in a dedicated Mobile Satellite Emergency Department (MSED) located outside the ambulance entrance of the normal Emergency Department.

This communication will discuss details and considerations of staffing, supply, patient flow, and the unique Mobile Satellite Emergency Department.

## 2. Materials and Methods

The Mobile Satellite Emergency Department (MSED) is part of a unique fleet of mobile medical assets originally developed by Hackensack University Medical Center, Hackensack, New Jersey, with federal government funding, circa 2010. The intent of the program was to provide a capability for an individual hospital to deploy mobile rapid response hospital-level medical care. Three prototype medical units and support vehicles were designed and built collaborating with vehicle manufacturers. There were also various research efforts related to the fleet development [1,2].

The prototype program and associated research projects were successful. When federal government funding ended a non-profit consortium of multiple hospitals and health care organizations were formed to continue the program. The program is known as the the Advanced Mobile Emergency Resource Coalition (AMERCO); http://www.amercousa.org/, accessed on 26 May 2020.

Five categories of missions were anticipated, including disaster response where local resources were overwhelmed or transport to them impaired, providing surge capacity during times of high patient volume, community outreach to the underserved, providing alternate clinical space during hospital repairs or construction and a clinical treatment area for patients requiring special isolation precautions. Over the years, the units have been deployed for all of these missions, including, in 2018, to St Croix, USVI, post-hurricane.

The units are nominally 50 foot trailers with expandable sides that are compatible with commercial tractors. They are highly mobile and are driven as standard tractor trailers. When open, the nominal footprint is 25 × 50 feet. Two are configured to function as well-equipped Emergency Departments or Intensive Care units. One is configured as a formal operating suite, complete with a hospital grade sterilizer. Units can function individually or be connected with enclosed climate controlled connectors.

If necessary, all three units can function alone, with no external utilities required. Each has an on-board electrical power generator, heating-ventilation-air conditioning (HVAC) systems, water tank, wastewater tank, oxygen supply, satellite and cellular communications links. During extended deployments, the units can be connected to external sources; detailed requirements for this are listed in Appendix A.

The MSED deployed for COVID-19 monoclonal antibody treatment was one of the Emergency Department/Intensive Care Units. The unit is configured with seven stretchers with bedside cardiorespiratory monitors connected to a central monitoring/alarm station, portable digital X-Ray, medical gas and suction, surgical procedure lighting, intravenous administration pumps and controllers, point of care blood gas and general laboratory analysis units, monitor/defibrillator/pacers, Omnicell medication distribution unit [3], code carts, supply carts, privacy curtains, multifunction stretchers, and a hydraulic platform lift for stretchers, wheelchairs, personnel, or supplies. It also includes a bathroom, sinks, refrigerator, macerator for disposable bedpans, and separate room for storage. Not all of these features were needed for the COVID monoclonal antibody treatment deployment.

Figure 1 is an external photograph and Figure 2 is an internal photograph of the MSED.

For this COVID-related deployment, the on-board HVAC systems were supplemented with two standalone HEPA recirculating filter units, one from Abatement Technologies (model HC800FD), and one from Dri-Eaz Defendair (model HEPA 500).

The MSED was located in close proximity to the Emergency Department ambulance entrance, which facilitated movement of patients, staff, and supplies and also allowed for rapid transfer of patients into the main Emergency Department if required. A parking deck is attached to the hospital building at that location, providing shelter from inclement weather.

Since this was an extended deployment, connections for power, water, wastewater, and computer network were made to the hospital building. The computer network connection allowed the host hospital to utilize their own VoIP telephones, Electronic Medical Record, and computer applications as if they were in the building [4].

Because the MSED emulated the in-house Emergency Department, only minimal staff training was required. This consisted of a walk-through orientation of the space, a review of the patient selection process for the MSED, as well as discussion regarding managing and communicating should a patient deteriorate or change clinical status. All key hospital stakeholders were provided with frequent updates on the daily hospital safety conference call.

## 3. Discussion and Results

Starting in December 2020, the mobile satellite emergency department treated adult pre-selected patients to receive the intravenous infusion Bamlanivimab, a monoclonal antibody provided to known positive COVID-19 patients. Patients meeting eligibility criteria for this emergency use authorization of medication were scheduled for outpatient infusion times one day in advance of their treatment dates. Selection criteria was the same as for patients treated in the in-building Emergency Department, as described in Appendix B.

Patients scheduled for Bamlanivimab infusions present to the emergency department, receive a quick patient registration, are examined by ED clinical staff (Physician, Physician Assistant, or Nurse Practionier) for medical stability, and are expeditiously escorted to the mobile satellite emergency department adjacent to the main emergency department building. Patients are physically distanced while maintaining appropriate face masking and grouped with similar patients. Although the MSED has the capability to treat seven patients simultaneously, being mindful of physical distancing requirements, the determination was made to treat no more than four Bamlanivimab infusion patients at one time. The assignment of four patients in the MSED also met the needs of the nursing team; the MSED was staffed with one registered nurse and one patient care technician. Two ‘rounds’ of patient schedules were maintained in the MSED; patients arriving at 07:00 for 08:00 infusions and patients arriving at 12:00 for 13:00 infusions. The rationale for having patients arrive 1 hour prior to their scheduled infusion time was to allow for the required pharmacy preparation time for the infusion. To ensure that Bamlanivimab infusions were not wasted the facility did not begin medication preparation until the patient had arrived on-site.

Treatment with Bamlanivimab requires approximately 1 hour of pharmacy preparation time in addition to 1 hour of intravenous infusion time. This is in addition to time required for registration, evaluations, and discharge.

The MSED proved to be a reliable, efficient, safe, and flexible treatment area that team members found to be a comfortable work environment. Although formal data had not been collected to verify team member satisfaction in the MSED, we believe that the absence of scheduling issues (inability to schedule team members, call-outs), as well as real-time feedback from the nursing and physician teams demonstrated team member satisfaction. Additionally, and importantly, team member scheduling for the MSED was on a voluntary basis. The emergency department leadership team had minimal if any challenges, staffing this ancillary care area.

During the first four complete months of 2021, 250 Bamlanivimab infusions were administered in the MSED. Patients triaged to the MSED had a completed nursing assessment and triage in 10 minutes (median). An evaluation by a physician/licensed independent provider was completed within 6 minutes. There were no safety issues identified during this period. Three patients were admitted to the main hospital from the MSED due to COVID-19 complications and worsening of their condition. The three patients requiring hospitalization were managed in the MSED until an inpatient bed became available. There were no issues or concerns with the MSED facility in caring for them.

Patients generally appeared to be fascinated and accepting of the MSED’s novel treatment environment.

Relative to patient throughput, for COVID-19 patients treated in the MSED with Bamlanivimab, the overall hospital arrival to departure length of stay for these patients was 199 minutes (median). This is illustrated in Figure 3 and Table 1.

This is in comparison to a hospital length of stay of 195 minutes (median) for patients of similar complexity patients being treated in the Emergency Department, as illustrated in Figure 4 and Table 2.

## 4. Conclusions

Although the primary function and purpose of the MSED during wave two of the pandemic was for outpatient monoclonal antibody infusions, the Emergency Department and hospital administration remained cognizant that at any time the function of the MSED may change, dependent on the catchment area’s healthcare needs. At any given moment, the ED team was prepared to utilize the MSED as a Fast Track area, a COVID-19 isolation area, or, for more critical patients, COVID or non-COVID. Although it was not necessary, discussion had taken place with the team so that the MSED was prepared to be utilized to house admitted patients. The ED team was also prepared to utilize the MSED to manage cardiac and other critical care patients, and we were confident that the MSED had the capabilities to do so. This flexibility allowed for strategic planning and preparation for whatever COVID-19 brought to the front doors of the Emergency Department.

The overall admission to discharge time for patients treated in the MSED with Bamlanivimab and Emergency Department patients of similar complexity treated in the hospital building was essentially the same, 199 minutes vs. 195 minutes, respectively, illustrating that alternate use care areas are an effective valued strategy when managing emergency department surge and/or pandemic occurrences.

Utilizing the MSED for the purpose of Bamlanivimab infusion resulted in making 829 hours of Emergency Department stretcher time available for patient use. However, we would like to mention that, in this ED patient processing, quality improvement activity hereby communicated that the authors are aware that the separate location for the 250 patients seen in the MSED vs. the 6290 seen in the main ED might have been influenced by several variables, including how busy the main ED was, the time of day, and staffing numbers, etc. We report on a retrospective, observational quality improvement activity.

The pandemic presented emergency departments with new and concerning challenges. The infectious nature and unknowns associated with the coronavirus demanded that ED’s function under new guidelines. The MSED offered an opportunity to flex the delivery of patient care services in an ever-changing environment. COVID-19 presented unique opportunities to improve patient care services and reinforced the continuing need for EDs to look to ancillary care areas for flexibility during time of patient care surge when Emergency Departments are stretched to capacity.

## Figures and Tables

**Figure 1 healthcare-09-00990-f001:**
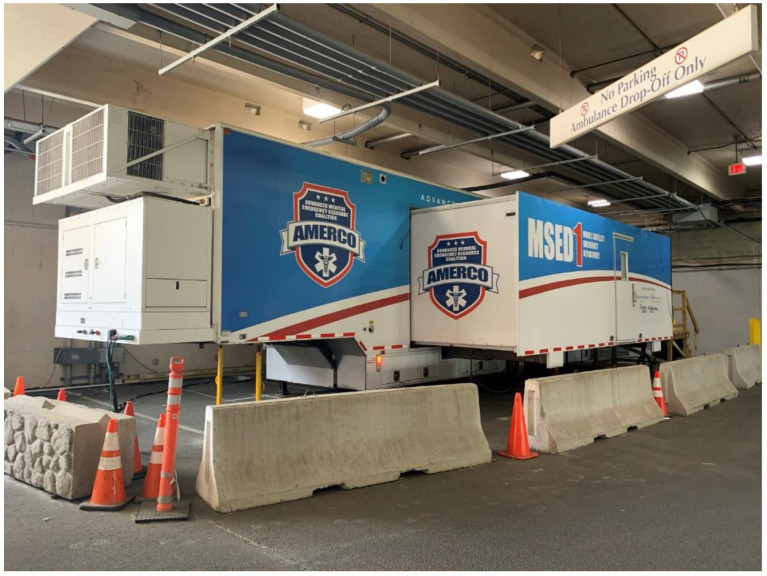
External view MSED outside the Emergency Department. Jersey Shore University Medical Center. Neptune, NJ, USA.

**Figure 2 healthcare-09-00990-f002:**
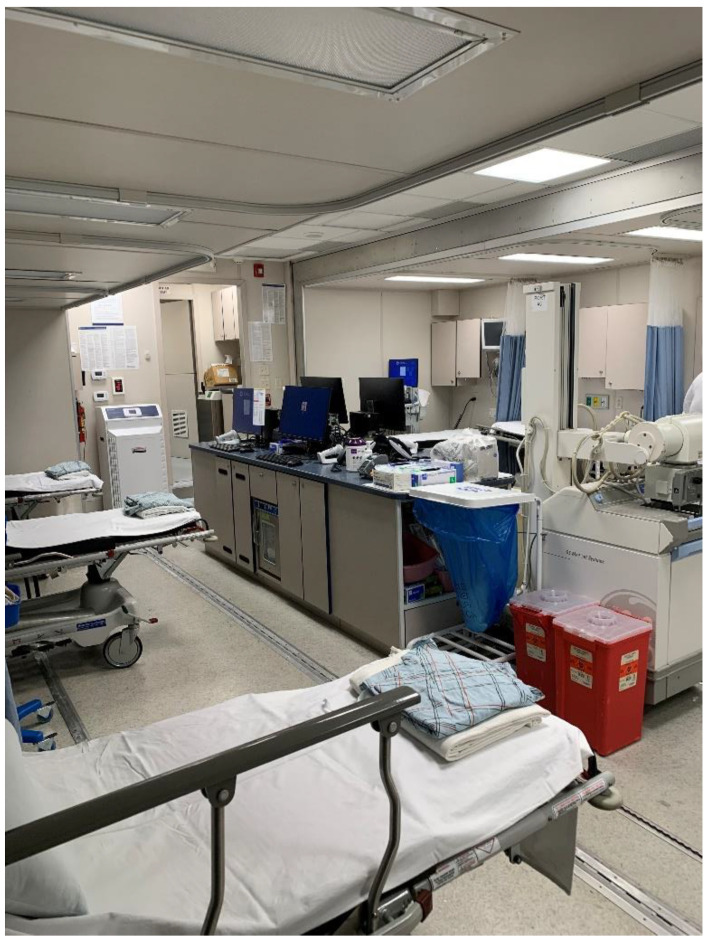
Internal view MSED set up for COVID treatment.

**Figure 3 healthcare-09-00990-f003:**
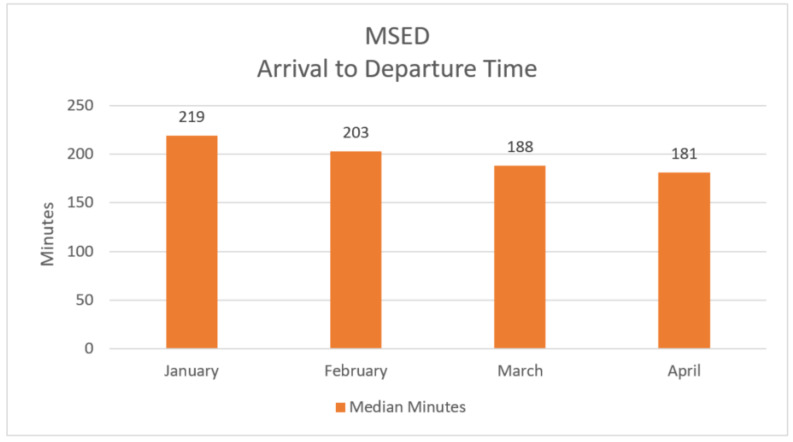
MSED patients arrival to departure time, first quarter, 2021.

**Figure 4 healthcare-09-00990-f004:**
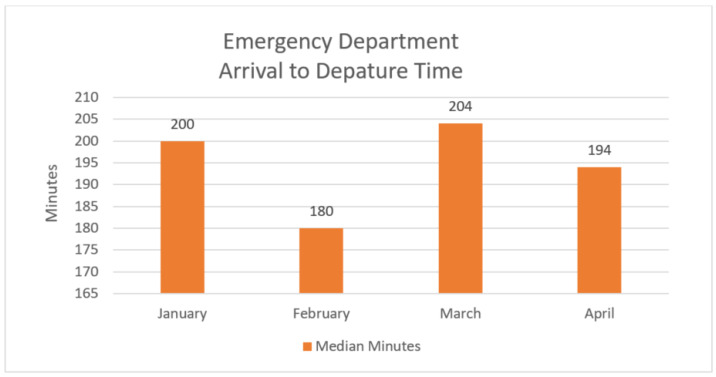
Emergency Department arrival to departure time, first quarter, 2021.

**Table 1 healthcare-09-00990-t001:** MSED patients count and arrival to departure time.

Arrival Month 2021	MSED Patients (Count)	MSED Patient Arrival to Departure Time (Median Minutes)
January	55	219
February	72	203
March	77	188
April	46	181
**Total**	**250**	**199**

**Table 2 healthcare-09-00990-t002:** Emergency Department patients count and arrival to departure time.

Arrival Month 2021	Emergency Department Patients (Count)	Emergency Department Patients Arrival to Departure Time (Median Minutes)
January	1552	200
February	1322	180
March	1647	204
April	1769	194
**Total**	**6290**	**195**

## Data Availability

The volume and length of stay data is maintained in a secure Hackensack Meridian Health system-wide volume, quality, and financial data base to health care institution standards. Data can be, and is, collected and accessed real time or retrospectively. Length of stay data is interfaced to this database directly from the secure patient medical record.

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
