# Peer review of "Use of a Unique Mobile Medical Asset in COVID Monoclonal Antibody Treatment"

_healthcare, 2021, doi:10.3390/healthcare9080990_

Round 1

Reviewer 1 Report

The article in word is attached, some comments/questions/thoughts.

Author Response

Reviewer 1 Comment Response

Thank  you very much for reviewing our paper.  Our response to each comment is attached below in blue text. 

Line 37  comment A1:    Is the MSED Unique or its use ?

Both the MSED and this application for it are unique.  The MSED's are one of a kind assets, funded as  prototypes by the federal government. This is explained in the Materials and Methods section 

Line 44 comment A2:     Is the 20,000 per year pediatric an exact number or an estimate

Both the 83,000 and 20,000 are approximate volumes

Line 56 comment a3:     Is the trailer unique or its use in this scenario ?

Both the MSED and this application for it are unique.  The MSED's are one of a kind assets, funded as prototypes by the federal government. This is explained in the Materials and Methods section 

Line 60 comment A4:     Why is this site/venue better than a space in the facility that is not used or underutilized (due to COVID or other                                               considerations)

Ideally it certainly would be best to treat all patients in the hospital building.  During the COVID crisis however  clinical space was at a premium due to patient volume, acuity, and isolation requirements. Hospitals were doing  things  like converting cafeteria dining areas into clinical space, etc.

Line 80 comment A5:    Check link (to AMERCO web site)

The link works from my computer, please let us know if it is not working for you  https://www.anercousa.org/

Line 89 comment A6:     Is this significant (MSED outside dimensions) ?

 Yes, the outside dimensions are important to give readers a feel for the amount of physical space required. We us the term "nominal" to indicate the numbers are approximate as opposed to being exact for design purposes. 

Line 95 comment A7:      How long can the unit be operated stand alone, based on fuel ?

The units can operate stand alone (without connection to external utilities) indefinitely as long as periodically resupplied with diesel fuel, water, and have their waste tank pumped out.  The MSED carries 500 gallons of  diesel fuel which typically can supply the onboard generator for several days.  There is variability of course depending outside temperature (HVAC energy requirement),clin ical use pattern, etc. If fully stand alone for extended periods  the on board water and wastewater tanks are supplemented  with external tanks to reduce re-supply and pump out requirements. 

Line 135 comment A8:    Who screens the patients ? nursing or clinicians, who is responsible for the patient while in the MSED ?

Patients receive a medical screening examination from a Physician, Physician  Assistant, or Nurse Practitioner.  A registered nurse is stationed in the MSED  with physician backup close by in the Emergency Department if needed

Line 138 comment A9:    Not sure I understand the "cohort" issue, are not all these patients COVID positive 

Yes, that is correct, they are all COVID positive.  We were just trying to re-emphasize  that point

Line 150 comment A10:  One-hour (not one hour ?)

Acknowledged

Reviewer 2 Report

Vey comprehensive report.  However I think you need to include a brief discussion of bamlanivimab  and moniclonal antbody therapy.  Who can receive this therapy and why.  Additionally, as patients age 12 and older are eligible, did you have any pediatric patients?  Finally in the discussion do you have any suggestions for use of this area for the ED in the future. 

Question: patients whose condition worsened were kept in this area until admission.  Any concerns about the increased level of care needed in termsof staffing and use of the area beyond the 2-3 hour expectation?

Again very comprehensive and well written.

Author Response

Response to Reviewer 2 comments:

Thank you very much for reviewing our paper.  Our response to your comments is in blue text below:

Very comprehensive report.  However I think you need to include a brief discussion of bamlanivimab  and moniclonal antbody therapy.  Who can receive this therapy and why.  Additionally, as patients age 12 and older are eligible, did you have any pediatric patients?  Finally in the discussion do you have any suggestions for use of this area for the ED in the future.

The paper assumes the reader is generally familiar with monoclonal antibody therapy for COVID.  As opposed to a discussion of the therapy itself our intent was to describe a unique physical space for monoclonal antibody treatment which frees up clinical space in the Emergency Department for treating other patients.We did not treat pediatric patients in the MSED

Yes, the MSED's have been used for several different types of missions in the past and their continued use is anticipated.  This is described in the the Materials and Methods section, line 82.  Missions have included providing medical care at the 2014 Superbowl, post hurricane emergency care, providing clinical space during hospital repair/construction, and  community outreach for veterans. 

Question: patients whose condition worsened were kept in this area until admission.  Any concerns about the increased level of care needed in terms of staffing and use of the area beyond the 2-3 hour expectation?

There were no issues or concerns in caring for these patients as discussed  in the Discussions and Results section, line 167.  The MSED was located about 100 feet from the Emergency Department ambulance entrance and is equipped with a hydraulic platform lift for stretchers so patients could be expeditiously moved to the in house Emergency Department if required.   

Reviewer 3 Report

The authors present a novel approach (Mobile Satellite Emergency Department) for treating COVID patients with monoclonal antibody Bamlanivimab. Below are a few considerations/ questions for revising and enhancing the presentation of these novel findings.  

Materials & Methods

  • Lines 109-110 mention that some of the features noted previously were not necessary for the treatment for which this manuscript is focused on - COVID monoclonal antibody - if that is the case, perhaps the authors could removing the "unnecessary" descriptions to not confuse the reader. 
  • Is it possible to include more information on how patients were selected to be treated in the MSED in this section? Also, did patients have to provide consent to be moved or treated in the MSED? 
  • Also, were healthcare staff given any unique training for working in the MSED? If so, brief details on the training provided would be helpful. If no training was necessary, then rationale would be helpful. 

Discussion & Results

  • How were patients pre-selected? What was the selection/eligibility criteria? As noted in the inquiry for the Methods section, those details would be best discussed in the Methods section.
  • Lines 128-152 seem to fit the Methods section. 
  • Figure 1 seems redundant since Table 1 has the same information - with the only difference being that Table includes the patient count. Similar comments apply to Figure 2 and Table 2. 

Conclusion

  • Based on the experience of using MSED for a pandemic, do the authors believe that MSEDs should be used at all hospitals during other potentially emergent situations only or should MSED usage be considered for general ED patient care services? Essentially, perhaps the authors could provide more information on the potential benefits/ implications of using MSEDs in the future as part of patient care services.

Author Response

Reviewer 3 Comment Response

Thank you very much for reviewing our paper.  Our response to your comments is in blue text attached below

Materials & Methods
Lines 109-110 mention that some of the features noted previously were not necessary for the treatment for which this manuscript is focused on - COVID monoclonal antibody - if that is the case, perhaps the authors could removing the "unnecessary" descriptions to not confuse the reader. 

We feel it is important to describe the full capability of the MSED since it is a unique asset.  The advanced care capability of the MSED is important in that the COVID patients being treated with monoclonal antibodies could deteriorate and require those advanced capabilities such as cardio-respiratory monitoring, etc. 

Is it possible to include more information on how patients were selected to be treated in the MSED in this section? Also, did patients have to provide consent to be moved or treated in the MSED? 

The MSED was treated as just an extension of the in house Emergency Department. Use was determined by staff availability, selection criterion was that same as for any COVID monoclonal antibody treatment candidates.  Patients did not need to give consent to be treated in the MSED, it is considered to be another clinical space, just as other locations in the in house Emergency Department. 

Also, were healthcare staff given any unique training for working in the MSED? If so, brief details on the training provided would be helpful. If no training was necessary, then rationale would be helpful. 

Minimal staff training was needed for working in the MSED since it's design and layout are essentially equivalent to an in house Emergency Department.  The drug distribution system (Omnicell),  Information Technology (IT) systems, Electronic Medical Record (EPIC), VoIP phone system, etc. are the same and are connected to the in building Emergency Department. 

The staff simply required a walk-through orientation of the space, a review of the patient selection process for the MSED as well as discussion regarding managing and communicating should a patient deterioration or change in clinical status been noted.  All key hospital stakeholders were provided frequent updates on the daily hospital safety conference call.

Discussion & Results
How were patients pre-selected? What was the selection/eligibility criteria? As noted in the inquiry for the Methods section, those details would be best discussed in the Methods section.
Lines 128-152 seem to fit the Methods section. 

Criteria for selecting  patients  for monoclonal treatment in the MSED was the same as for in the in house Emergency Department. 

Figure 1 seems redundant since Table 1 has the same information - with the only difference being that Table includes the patient count. Similar comments apply to Figure 2 and Table 2. 

Yes, that is correct.  We feel it is helpful to include both the figure and table though to help illustrate the conclusions

Conclusion
Based on the experience of using MSED for a pandemic, do the authors believe that MSEDs should be used at all hospitals during other potentially emergent situations only or should MSED usage be considered for general ED patient care services? Essentially, perhaps the authors could provide more information on the potential benefits/ implications of using MSEDs in the future as part of patient care services.

The MSED program is well established for a number of years with a proven track record for both pandemic and non-pandemic use.  The five categories of missions include disaster response where local resources are overwhelmed, surge capacity during periods of high patient volume,  community outreach to the undeserved, providing clinical space during hospital repairs or construction, and providing clinical space for patients requiring special isolation (such as Ebola).  This is described in the Materials and Methods section